# Anticancer Effect of Benzimidazole Derivatives, Especially Mebendazole, on Triple-Negative Breast Cancer (TNBC) and Radiotherapy-Resistant TNBC In Vivo and In Vitro

**DOI:** 10.3390/molecules26175118

**Published:** 2021-08-24

**Authors:** Hoon Sik Choi, Young Shin Ko, Hana Jin, Ki Mun Kang, In Bong Ha, Hojin Jeong, Haa-Na Song, Hye Jung Kim, Bae Kwon Jeong

**Affiliations:** 1Department of Radiation Oncology, Gyeongsang National University Changwon Hospital, Gyeongsang National University College of Medicine, Jinju 52727, Korea; radignuh@gnuh.co.kr (H.S.C.); jsk92@gnu.ac.kr (K.M.K.); 2Institute of Health Science, Gyeongsang National University, Jinju 52727, Korea; shini33@naver.com (Y.S.K.); hahaha-_-0001@daum.net (H.J.); nicehib@gnuh.co.kr (I.B.H.); jeong3023@gnu.ac.kr (H.J.); songhaana@gnu.ac.kr (H.-N.S.); 3Biomedical Research Institute, Gyeongsang National University Hospital, Jinju 52727, Korea; 4Department of Pharmacology, Gyeongsang National University College of Medicine, Jinju 52727, Korea; 5Department of Radiation Oncology, Gyeongsang National University Hospital, Gyeongsang National University College of Medicine, Jinju 52727, Korea; 6Division of Hemato-Oncology, Department of Internal Medicine, Gyeongsang National University Hospital, Gyeongsang National University College of Medicine, Jinju 52727, Korea

**Keywords:** triple-negative breast cancer, anthelmintic, benzimidazole, mebendazole, cancer stem cell, radioresistance

## Abstract

In this study, we aimed to evaluate the anticancer effect of benzimidazole derivatives on triple-negative breast cancer (TNBC) and investigate its underlying mechanism of action. Several types of cancer and normal breast cells including MDA-MB-231, radiotherapy-resistant (RT-R) MDA-MB-231, and allograft mice were treated with six benzimidazole derivatives including mebendazole (MBZ). Cells were analyzed for viability, colony formation, scratch wound healing, Matrigel invasion, cell cycle, tubulin polymerization, and protein expression by using Western blotting. In mice, liver and kidney toxicity, changes in body weight and tumor volume, and incidence of lung metastasis were analyzed. Our study showed that MBZ significantly induced DNA damage, cell cycle arrest, and downregulation of cancer stem cell markers CD44 and OCT3/4, and cancer progression-related ESM-1 protein expression in TNBC and RT-R-TNBC cells. In conclusion, MBZ has the potential to be an effective anticancer agent that can overcome treatment resistance in TNBC.

## 1. Introduction

According to the World Health Organization (WHO) data for 2019, cancer is a major public health and economic problem and is a leading cause of death worldwide [1]. Global cancer incidence was 19.3 million in 2020. Breast cancer is the leading cause of cancer with 2.3 million new cases, accounting for 11.7% of all cancer cases. It is the fifth leading cause of cancer-related deaths worldwide, with 685,000 deaths [2]. At the 2013 St. Gallen International Breast Cancer Conference [3], breast cancer was classified into the following molecular subtypes: luminal A, luminal B, human epidermal growth factor receptor 2 (HER2) enhanced, basal-like triple-negative breast cancer (TNBC), and others. These clinical classifications are used to determine the management policy and predict the prognosis of breast cancer patients. TNBC is defined by the lack of expression of estrogen receptor (ER), progesterone receptor (PR), and HER2, which accounts for approximately 12–17% of all breast cancers [4]. TNBC, which is less sensitive to hormonal therapy and HER2-targeted therapy compared to other subtypes of breast cancer, has no specific treatment guidelines. In addition, the cancer cells of TNBC have self-renewal and regenerating characteristics, such as cancer stem cells, based on the expression of cell-surface markers CD44+CD24− and aldehyde dehydrogenase 1 (ALDH1) [5]. Therefore, most TNBC patients receive a combination of treatment, surgery, postoperative chemotherapy, and postoperative radiotherapy (RT) based on standard treatment algorithms [6]; however, they have a short survival period and mortality rate of 40% within 5 years from diagnosis [7] because of acquisition of therapy resistance. 

Therefore, it necessary to investigate more effective treatment modalities or targets for TNBC and therapy-resistant TNBC to overcome the resistance to traditional treatments and recurrence. However, the development of new drugs is costly and requires a long time of approximately 10–15 years. Furthermore, it is challenging owing to its low success rate. Therefore, there is a need for a new strategy to reduce the time and cost required for the identification of new drugs. One of these approaches is drug repurposing; that is, the strategy to invest in new uses of already approved or in-market drugs in addition to the original indications [8]. In addition to timesaving and cost reduction, drug repurposing has a significant advantage in ensuring safety through well-known pharmacokinetic and pharmacodynamic features. In the field of oncology, with the advent of metformin, immunotherapy, and alpha-lipoic acid, interest is increasing in drug repurposing of non-anticancer drugs to anticancer drugs [9,10].

In this study, benzimidazole derivatives were selected as drug repurposing agents for investigating their anticancer effects in TNBC. The reasons for examining benzimidazole derivatives as candidate agents include evidence that some anthelmintic mechanisms are associated with oncogenic pathways and that several previous cell lines or animal studies have reported anticancer effects [11,12,13,14]. The original indication of anthelmintics was to treat infections caused by parasites in the human intestine. 

Initially, they were developed to treat veterinary parasites and later used as clinical drugs for humans. Among them, benzimidazole derivatives have been widely used in human and veterinary medicine since the 1960s because of their low cost, high efficacy, and safety without serious side effects. Based on these considerations, we explored the anticancer effect and the mechanisms of benzimidazole derivatives such as albendazole (ALB), albendazole sulfoxide (ALB-sulfoxide), flubendazole (FLU), fen-bendazole (FBZ), mebendazole (MBZ), and oxfendazole (OFZ) against TNBC and RT-resistant (RT-R)-TNBC in vivo and in vitro to confirm the possibility of drug repurposing of benzimidazole derivatives.

## 2. Results

### 2.1. Benzimidazole Derivatives—Specifically Four Derivatives, ALB, FLU, FBZ, and MBZ—Significantly Reduced Colony Formation in MDA-MB-231 and RT-R-MDA-MB-231 Cells at No Cytotoxic Doses

First, we determined the effects of six benzimidazole derivatives (ALB, ALB-SUL, FLU, FBZ, MBZ, and OFZ) (Figure 1) at concentrations of 0.01, 0.1, 0.5, 1, 2, 5, and 10 μM) on the viability of the normal breast epithelial cell line MCF10A and TNBC cell line MDA-MB-231 and RT-R-MDA-MB-231 by the CCK assay. Regarding normal breast epithelial cells, MCF10A, none of the drugs affected the cell viability at doses 0.01, 0.1, 0.5, and 1 μM for 24–72 h. Even ALB-SUL and OFZ did not show any toxicity up to 10 μM after 72 h of incubation. However, a high dose (10 μM) of ALB, FLU, FBZ, and MBZ decreased the cell viability of MCF10A by approximately 40–50% compared to the control at 72 h (Figure 2a). Regarding TNBC cells, for MDA-MB-231 and RT-R-MDA-MB-231 cells, the six derivatives hardly affected cell viability at doses of 0.01, 0.1, 0.5, 1, 2, 5, and 10 μM for 24 h. However, when incubated for 72 h after 10 μM of ALB, FLU, FBZ, or MBZ administration, cell viability was slightly decreased, by approximately 35–50% compared to the control (Figure 2b). Then we examined the effect of benzimidazole derivatives on the colony-forming ability of MDA-MB-231 and RT-R-MDA-MB-231 cells. Following treatment with each of the six benzimidazole derivatives at the indicated doses for 24 h, the cells were changed to complete fresh media and incubated without the drugs for an additional 10 days. All drugs have anticancer effects by reducing clonogenicity from 0.1 μM, which did not affect cell viability for 24–72 h in both MDA-MB-231 and RT-R-MDA-MB-231 cells. Four benzimidazole derivatives, ALB, FLU, FBZ, and MBZ, showed a significant inhibitory effect on the formation of colonies by RT-R-MDA-MB-231 cells from 0.1 μM (Figure 3). These effects were also observed in other non-TNBC cell lines, such as MCF-7 and T47D, and their corresponding RT-R-MCF-7 and RT-R-T47D cells (data not shown). Therefore, we selected four drugs (ALB, FLU, FBZ, and MBZ) that significantly inhibit colony-forming ability in the TNBC cell line for subsequent experiments at 0.1–1 μM.

### 2.2. ALB, FLU, FBZ, and MBZ Exhibit Anticancer Effects in an Allograft Mouse Model without Weight Loss or Hepatic and Kidney Toxicity

We then examined whether the benzimidazole derivatives ALB, FLU, FBZ, and MBZ have anticancer effects on an in vivo mouse model. To determine the safety of the drugs at 10 mg/kg, the drugs were orally administered once a day on a schedule of 5-days-on and 2-days-off for 2 weeks. Figure 4 shows that ALB, FLU, FBZ, and MBZ did not induce any hepatic damage, as assayed by determination of plasma ALT and AST levels (Figure 4a,b); or kidney toxicity, as assayed by the determination of plasma creatinine levels (Figure 4c); or body weight loss (Figure 4d). The four drugs were administered to the 4T1-injected mice, and they showed no changes in body weight (Figure 4e). ALB, FLU, FBZ, and MBZ decreased tumor volume in the 4T1-injected allograft mice model (Figure 4f), and a significant decrease in tumor volume was observed in FBZ- and MBZ-treated mice 28 days after administration (Figure 4g,h). FLU- and MBZ-treated 4T1-injected mice showed significantly decreased lung metastasis (Figure 4i). In the RT-R-4T1-injected mouse model, FBZ and MBZ significantly decreased tumor volume (Figure 4k–m), but only MBZ showed an inhibitory effect on lung metastasis (Figure 4n) without changes in body weight (Figure 4j).

### 2.3. MBZ Effectively Suppresses Cell Migration and Invasion at Low Doses

Because the in vivo animal study showed that of six benzimidazole derivatives tested, MBZ was most effective in decreasing tumor volume and lung metastasis without causing any toxicity, we further investigated the mechanisms underlying the anticancer effects of MBZ. Scratch-wounded MDA-MB-231 and RT-R-MDA-MB-231 cells almost closed the wound after 42 h, which was significantly inhibited by MBZ at 0.5 and 1 μM (Figure 5a,b). Additionally, MBZ significantly reduced the invasion of both MDA-MB-231 and RT-R-MDA-MB-231 cells through the EC-Matrigel-coated insert well membrane (Figure 5c,d) at low doses of 0.5 and 1 μM.

### 2.4. MBZ Induces Cell Cycle Arrest in Both MDA-MB-231 and RT-R-MDA-MB-231 Cells in the G2/M Phase by Inhibiting Polymerization of Tubulin, and Effectively Induces DNA Damage in RT-R-MDA-MB-231 Cells

We then determined the effect of MBZ on cell cycle progression. Treatment of MDA-MB-231 and RT-R-MDA-MB-231 cells with MBZ for 24 h arrested the cell cycle in a dose-dependent manner at the G2/M phase, showing a reduction in the number of cells in the G0-G1 phase and parallel induction of G2/M phase. MBZ did not affect the S phase of MDA-MB-231 cells, which was rather increased at 0.5 μM, but significantly reduced the S phase of RT-R-MDA-MB-231 cells at 0.5 and 1 μM (Figure 6a). Next, we examined whether MBZ inhibited tubulin polymerization. Results showed that treatment of cells with MBZ for 24 h did not change tubulin levels in the supernatant but significantly decreased the levels of tubulin in the pellet at 0.5 and 1 μM (Figure 6b), suggesting MBZ-mediated suppression of tubulin polymerization. Moreover, MBZ increased the expression of cyclin B1 at 0.5 and 1 μM but not that of cyclin D1 after treatment for 24 h (Figure 6c). A time-dependent experiment with MBZ at 0.5 μM showed that the expression of cyclin B1 by MBZ was significantly induced 6 h after treatment and was maintained until 24 h, and then decreased below the control levels at 48 h (Figure 6d). We also investigated whether MBZ damages DNA in both MDA-MB-231 and RT-R-MDA-MB-231 cells. As shown in Figure 6e, MBZ significantly induced cleaved caspase-3 levels at 0.5 and 1 μM in both MDA-MB-231 and RT-R-MDA-MB-231 cells, where the levels of cleaved caspase-3 were higher in MDA-MB-231 than in RT-R-MDA-MB-231 cells in the control and the MBZ-treated groups. In addition, MBZ induced DNA damage when it was observed with phosphorylated γH2AX (pH2AX), a marker of DNA double-strand break in MDA-MB-231 cells, showing a significant induction at 0.5 and 1 μM. RT-R-MDA-MB-231 cells showed a more increased pH2AX level in response to MBZ (Figure 6f) in contrast to cleaved caspase-3. Densitometry readings/intensity ratio and whole blot for Figure 6 are presented in Appendix A.

### 2.5. MBZ Downregulates the Expression of Cancer Stem Cell (CSC) Markers CD44 and OCT3/4 and Cancer-Progression-Related Protein Endothelial Cell-Specific Molecule-1 (ESM-1)

Lastly, we examined the effect of MBZ on CSC markers and the cancer-progression-related protein ESM-1, which was reported in a previous study [15]. CSC markers CD44 and OCT3/4, and ESM-1 were effectively downregulated following incubation with 0.5 and 1 μM MBZ (Figure 7a–c). There was no difference in the inhibitory effect of 0.5 and 1 μM MBZ on CD44 and OCT3/4; however, the inhibitory effect on ESM-1 was dose-dependent. Densitometry readings/intensity ratio and whole blot for Figure 7 are presented in Appendix A.

## 3. Discussion

TNBC is the worst of all the breast cancer subtypes, with limited treatment options due to its nature, and poor prognosis [7]. Moreover, the radioresistance of TNBC cells remains a fundamental barrier to the maximum efficacy of RT, and therefore, there is an increased rate of recurrence and treatment failure, ultimately resulting in death. Therefore, various treatment strategies, including the development of new drugs such as immunotherapeutic agents or new therapeutic targets, are being investigated to improve TNBC prognosis and to overcome therapy resistance [16,17]. We also analyzed the applicability of benzimidazole derivatives in terms of drug efficacy and safety using in vivo and in vitro models of TNBC and RT-R-TNBC and investigated their mechanism of action in the treatment of TNBC and RT-R-TNBC.

Our findings demonstrated that the four benzimidazole derivatives ALB, FLU, FBZ, and MBZ had anticancer effects, such as reduced colony formation in MDA-MB-231 cells, a TNBC cell line, and RT-R-MDA-MB-231 cells. In particular, MBZ showed the strongest anticancer effect and reduction of lung metastasis in the mouse model; therefore, we further investigated the mechanisms by which MBZ exerts anticancer effects in MDA-MB-231 and RT-R-MDA-MB-231 cells. MBZ induced cell cycle arrest at the G2/M phase and increased cyclin B1 levels, but not those of cyclin D1, for up to 24 h after treatment, and decreased these levels thereafter. In addition, tubulin polymerization was inhibited in the MBZ-treated group. These results suggest that MBZ promotes cell cycle progression from G1 to G2-M by increasing cyclin B1 protein levels and then arresting cells at the M phase by inhibiting tubulin polymerization at 24 h. α- and β-tubulins polymerize into microtubules, major constituents of mitotic spindles, and function in many essential processes, including cell division [18]. Drugs that inhibit tubulin polymerization or promote microtubule depolymerization cause mitotic arrest [19,20]. Tubulin-binding drugs such as paclitaxel, colchicine, and vincristine kill cancerous cells by inhibiting microtubule dynamics, which are required for DNA segregation and, therefore, for cell division [21]. Benzimidazole is a heterocyclic aromatic organic compound, which consists of the fusion of benzene and imidazole. Benzimidazole derivatives also act as anthelmintics by binding to tubulin, consistent with this study showing anticancer effects [22,23]. 

In addition, MBZ caused DNA damage in MDA-MB-231 and RT-R-MDA-MB-231 cells, as reflected by decreased clonogenicity and tumor volume in allograft mice, and increased pH2AX and cleaved caspase-3 levels. In the case of cleaved caspase-3, RT-R-MDA-MB-231 cells showed decreased levels compared to MDA-MB-231 cells in the control, but this was significantly induced by MBZ (at 0.5 and 1 μM). 

TNBC cells show characteristics of CSCs, which have tumor-initiating potential and possess self-renewal ability, promoting treatment resistance, tumor formation, growth, migration, invasion, and metastasis, leading to poor prognosis [24]. Therefore, manipulation of CSCs in TNBC is considered a target for therapeutic strategies. Bay et al. studied the effects of hypoxia-inducible factor-2α on MDA-MB-231 cells. They reported that downregulation of CD44, a marker for breast CSCs, is involved in inhibiting tumor stem cell self-renewal and increases the sensitivity of stem cells to RT and chemotherapy [25]. Another CSC marker, OCT 3/4, may also be a potential target in advanced breast cancer [26]. In this study, MBZ significantly downregulated the expression of CSC markers CD44 and OCT3/4 in MDA-MB-231 and RT-R-MDA-MB-231 cells. Even though we examined other CSC markers such as ALDH1 and Noth-4, no changes were shown in the MBZ-treated group (data not shown). The inhibition of cell migration and invasion in vitro and lung metastasis in a mouse model, as shown in this study’s results, is probably due to the downregulation of these CSC markers.

ESM-1, also known as endocan, is overexpressed in various cancers and has adverse effects [27]. ESM-1 plays an important role in tumor progression by regulating angiogenesis [28]. ESM-1 has been used as a potential prognostic marker in TNBC because of its overexpression, associated with significantly increased distant metastasis [29]. In addition, ESM-1 is overexpressed in RT-R-MDA-MB-231 cells [15]. In this study, MBZ inhibited ESM-1 protein in both MDA-MB-231 and RT-R-MDA-MB-231 cells, suggesting that ESM-1 could be a therapeutic target through which MBZ shows an anticancer effect in breast cancer.

Other researchers have also been very interested in drug repurposing of anthelmintics, and some of their studies have reported cytotoxic effects in breast cancer. Zhou et al. [30] studied the anticancer effect of FLU on TNBC in vitro and in vivo. They argued that FLU showed anti-proliferative and anti-migration effects on TNBC as a mechanism for increasing autophagic cell death, and that EVA1A was the main target in this process. In our study, the anticancer effects of six benzimidazole derivatives, including FLU, were simultaneously studied, and MBZ showed a greater anticancer effect than FLU in TNBC cells and RT-resistant TNBC cells. Vlashi et al. [31] focused on TNBC, which contains a larger portion of breast cancer-initiating cells (BCIC), having tumor-initiating potential, and evaluated whether MBZ overcame treatment resistance and showed a synergistic effect when combined with RT. They argued that MBZ could effectively eliminate the BCIC pool and prevent conversion of breast cancer cells to therapy-resistant BCIS caused by ionizing radiation, thereby improving tumor control when combined with RT. Their study is similar to our study, which confirmed the anticancer effect of MBZ on TNBC cells that acquired resistance after RT irradiation, although the study procedure is different, in that it seeks to overcome treatment resistance.

Many benzimidazole derivatives, including MBZ, have already been proven safe and commercialized as anthelmintic or fungicides. In this study, 0.5 and 1 μM doses of MBZ showed anticancer effects, and these doses did not reduce the viability of normal breast epithelial cells. Moreover, oral treatment with 10 mg/kg of MBZ daily for 2 consecutive weeks on a schedule of 5-days-on and 2-days-off did not affect body weight and did not cause liver or kidney toxicity in treated mice, but showed a significant decrease in tumor volume and lung metastasis. According to several reports, MBZ showed anticancer effect at various doses from 7.5 mg/kg to 200 mg/kg [31,32,33,34,35]. To test the anticancer effect of MBZ at safe dose ranges, we chose treatment with 10 mg/kg MBZ 5 days weekly for 2 weeks. Drug repurposing is a drug development approach that identifies new indications for drugs that have already been proven to be safe. This technology has the advantage of reducing costs and shortening the development period and is a trend that has recently been in the spotlight [36]. Further studies, including clinical trials to confirm the anticancer effect or mechanism, are essential. This study showed (1) the need for additional therapeutic strategies in TNBC and RT-R-TNBC, (2) the efficacy of MBZ as an anticancer agent, (3) various targets for action, and (4) the proven safety of MBZ. In view of these, we recommend the use of MBZ as an anticancer drug for TNBC and RT-R-TNBC as a new indication according to the drug repurposing concept.

## 4. Materials and Methods

### 4.1. Materials

ALB, ALB-SUL, FLU, FBZ, MBZ, and OFZ were purchased from Sigma-Aldrich (St. Louis, MO, USA). RPMI 1640 medium, fetal bovine serum (FBS), and antibiotics (penicillin/streptomycin) were obtained from HyClone Laboratories (South Logan, UT, USA). Anti-Cyclin D1, cyclin B1, CD44, and endothelial cell-specific molecule-1 (ESM-1) antibodies were provided by Abcam (Cambridge, UK). Anti-α-tubulin antibody and anti-OCT3/4 antibody were purchased from Cell Signaling Technology (Beverly, MA, USA) and Santa Cruz Biotechnology (Dallas, TX, USA), respectively. The BD Matrigel™ basement membrane matrix was supplied by BD Biosciences (San Diego, CA, USA). Enhanced chemiluminescence (ECL) Western blotting detection reagent was obtained from Bio-Rad (Hercules, CA, USA). All other chemicals were purchased from Sigma-Aldrich.

### 4.2. Cell Culture

The human breast cancer cell lines MDA-MB-231, MCF7, and T47D were obtained from the Korea Cell Line Bank (Seoul, Korea). The breast epithelial MCF10A and mouse breast cancer cell line 4T1 were obtained from the American Type Culture Collection (ATCC, Manassas, VA, USA). RT-R-MDA-MB-231 cells were generated by applying repetitive small doses of X-ray irradiation (2 Gy) until a final dose of 50 Gy was achieved. All cancer cell lines were cultured in RPMI 1640 supplemented with 10% FBS and 1% penicillin/streptomycin, and incubated at 37 °C in a humidified atmosphere of 5% CO_2_ and 95% air.

### 4.3. Cell Viability Assay

The cells were seeded in 24-well plates (1 × 104 cells/well). Cells were treated with six benzimidazole derivatives (ALB, ALB-SUL, FLU, FBZ, MBZ, and OFZ) at the indicated concentrations (0.01, 0.1, 0.5, 1, 2, 5, and 10 μM) for 24–72 h. After treatment, 10 μL CCK-8 reagent was added to the wells and incubated for 30 min at 37 °C in the dark. The optical density of each well was measured at 450 nm using a microplate reader (Molecular Devices VersaMax, Sunnyvale, CA, USA).

### 4.4. Colony Formation Assay

Cancer cells were seeded in 6-well plates (1 × 103 cells/well). The cells were treated with six benzimidazole derivatives at the indicated doses (0.1, 0.5, 1, 5, and 10 μM) at 37 °C for 24 h. After treatment, the culture medium was replaced with complete medium every 2–3 days. After 10 days, the medium was discarded, and the cells were washed with phosphate-buffered saline (PBS). The colonies were fixed in 100% methanol for 10 min at room temperature and stained with 0.1% Giemsa staining solution for 30 min at room temperature, and visible colonies were counted.

### 4.5. Scratch Wound Healing Assay

MDA-MB-231 and RT-R-MDA-MB-231 cells were cultured in 6-well plates and scratched with a sterile 200-μL pipette tip. The cells were then treated with four benzimidazole derivatives (ALB, FLU, FBZ, and MBZ) at the indicated concentrations (0.1, 0.5, and 1 μM). After 42 h, the cells were washed with PBS, and images were taken using an Olympus photomicroscope. The number of cells that migrated to the scratched area was counted.

### 4.6. Matrigel Invasion Assay

The upper chambers of the inserts were coated with 100 μL of Matrigel (1 mg/mL), and endothelial cells (EC) (2 × 105 cells) were added to the Matrigel-coated inserts. MDA-MB-231 and RT-R-MDA-MB-231 cells were treated with four benzimidazole derivatives (ALB, FLU, FBZ, and MBZ) at the indicated doses (0.1, 0.5, and 1 μM) for 24 h. The treated cells were collected and added to the upper chambers (2 × 105 cells/insert) in serum-free media, and 500 μL of RPMI-1640 with 10% FBS was added to the lower chambers. The invasion chambers were incubated overnight (16 h) in a cell culture incubator at 37 °C. The noninvasive cells that remained on the upper surface of the insert membranes were removed by scrubbing. The cells on the lower surface of the insert membranes were stained with DAPI, and the invading cells were counted under a fluorescence microscope (Eclipse Ti-U, Nikon, Tokyo, Japan).

### 4.7. Cell Cycle Analysis

MDA-MB-231 and RT-R-MDA-MB-231 cells were treated with ALB, FLU, FBZ, and MBZ at the indicated concentrations (0.1, 0.5, and 1 μM) for 24 h. The cells were harvested and washed with PBS. The harvested cells were fixed with 75% ethanol at 4 °C overnight, and then stained with PI solution (50 μg/mL of PI, 0.7 μg/mL RNase A) at room temperature for 30 min in the dark. DNA content was analyzed using a BD FACS II flow cytometer (BD Bioscience, San Jose, CA, USA).

### 4.8. Tubulin Polymerization Assay

MDA-MB-231 and RT-R-MDA-MB-231 cells were treated with ALB, FLU, FBZ, and MBZ at the indicated concentrations (0.1, 0.5, and 1 μM) for 24 h. After treatment, the cells were washed with PBS, harvested, and lysed in lysis buffer containing 20 mM Tris HCl (pH 6.8), 140 mM NaCl, 1 mM MgCl2, 2 mM EDTA, 0.5% NP40, and protease inhibitors. The samples were centrifuged at 16,000× *g* for 10 min at room temperature to separate soluble (S) tubulin. The supernatant containing depolymerized (S) tubulin was transferred to a new tube. The pellet containing polymerized (P) tubulin was re-suspended in RIPA buffer containing 50 mM Tris-HCl (pH 7.5), 150 mM NaCl, 1% NP-40, 0.1% SDS, 0.5% sodium deoxycholate, and protease inhibitors, and the samples were centrifuged at 16,000× *g* for 15 min at 4 °C. The supernatant containing P tubulin was transferred to a new tube. Equal amounts of protein were subjected to 8–10% sodium dodecyl sulfate–polyacrylamide gel electrophoresis (SDS-PAGE) and analyzed by using Western blotting with anti-α-tubulin antibody (2144S; 1:1000, Cell Signaling Technology).

### 4.9. Western Blot Analysis

Cells were harvested and lysed in a RIPA buffer, and centrifuged at 16,000× *g* for 20 min at 4 °C. The supernatants were collected to determine protein concentration using the Bradford method. Equal amounts of protein were subjected to 8–10% SDS-PAGE for 2 h at 100 V. The separated proteins were transferred onto Hybond-P+ polyvinylidene difluoride membranes (Amersham, Buckinghamshire, UK). The membranes were blocked with 5% nonfat milk in Tris-buffered saline containing 0.05% Tween-20 (TBS-T) for 1 h at room temperature, and then incubated with the following primary antibodies: anti-cyclin D1 (ab16663; 1:1000, Abcam), anti-cyclin B1 (ab32053; 1:1000, Abcam), anti-cleaved caspase-3 (9661S; 1:1000, Cell Signaling Technology), anti-pH2AX (2577S; 1:1000, Cell Signaling Technology), anti-CD44 (ab51037; 1:1000, Abcam), anti-ESM1 (ab103590; 1:1000, Abcam), and anti-OCT3/4 (sc-9081; 1:1000, Santa Cruz Biotechnology) antibodies. The bound antibodies were detected using horseradish peroxidase-conjugated secondary antibodies and an ECL (Bio-Rad) Western blotting detection reagent. The relative protein levels were normalized to those of β-actin (MA5–15739, Thermo Fisher Scientific, Waltham, MA, USA) used as a loading control.

### 4.10. Animal Experiments

Female athymic nude mice (6-week-old) were purchased from OrientBio (Gyeonggi-do, Korea). They were maintained under the following constant ambient conditions: 22 °C to 26 °C, 40% to 60% humidity, 12 h light/dark cycle, and free access to sterilized food and water. The mice were subcutaneously injected with 4T1 cells or RT-R-4T1 cells (5 × 10^4^ cells/100 μL). Seven days after injection, body weight and tumor volume were measured three times a week. When tumor volumes reached 100 mm^3^, the mice were divided into 10 groups (*n* = 7/each group): (1) mice injected with 4T1 cells (4T1 group), (2) 4T1 group + ALB, (3) 4T1 group + FLU, (4) 4T1 group + FBZ, (5) 4T1 group + MBZ, (6) mice injected with RT-R-4T1 cells (RT-R-4T1 group), (7) RT-R-4T1 group + ALB, (8) RT-R-4T1 group + FLU, (9) RT-R-4T1 group + FBZ, and (10) RT-R-4T1 group + MBZ. The drugs (ALB, FLU, FBZ, MBZ; 10 mg/kg/mL) suspended in 0.25% sodium carboxymethyl cellulose (CMC) solution were administered by oral gavage daily for 2 consecutive weeks on a schedule of 5-days-on and 2-days-off. The mice were sacrificed 28 days after injection, and tumor size and lung metastasis were measured. To determine the safety of drugs, the mice were divided into five groups (*n* = 6/each group): (1) control mice, (2) ALB-treated mice, (3) FLU-treated mice, (4) FBZ-treated mice, and (5) MBZ-treated mice. The drugs were administered by oral gavage daily for 2 consecutive weeks on a schedule of 5-days-on and 2-days-off. Body weights and tumor volumes were measured every 3 days. After 2 weeks, the mice were sacrificed, and their blood was collected. Plasma was separated from the blood sample by centrifugation at 3000× *g* for 15 min at 4 °C. Plasma alanine aminotransferase (ALT) and aspartate aminotransferase (AST) levels were measured by using assay kits from IVD Lab (Uiwang, Korea) and a spectrophotometer (Shimadzu UV-1800 spectrophotometer, Tokyo, Japan). Plasma creatinine was directly measured using the colorimetric Jaffe method. The animal experimental protocol was approved by the Institutional Animal Care and Use Committee of Gyeongsang National University (approval number: GNU-200603-M0030).

### 4.11. Statistical Analysis

All data were analyzed using GraphPad Prism 7 software, version 7.00 (GraphPad Software, San Diego, CA, USA). Scanning densitometry was performed using an Image Master^®^ VDS system (Pharmacia Biotech Inc., San Francisco, CA, USA). Statistical comparisons were made using unpaired Student’s t-test for two-group comparisons or one-way ANOVA followed by Tukey’s multiple comparisons test for comparisons of three or more groups. All results are presented as the mean ± standard deviation.

## 5. Conclusions

Taken together, our results suggest that MBZ has the strongest anticancer effect in both MDA-MB-231 and RT-R-MDA-MB-231 cells by blocking cell cycle progression at the G2/M phase. In addition, the anticancer effect of MBZ was through inhibition of the expression of CD44, OCT 3/4, and ESM-1, which appeared to be linked to RT resistance of RT-R MDA-MB-231 cells.

## Figures and Tables

**Figure 1 molecules-26-05118-f001:**
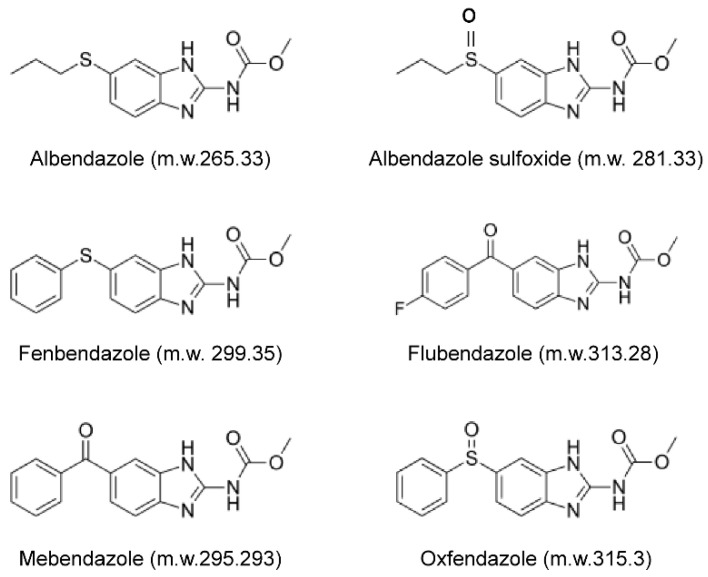
Structure of benzimidazole derivatives. Albendazole, ALB; ALB-sulfoxide, ALB-SUL; fenbendazole, FBZ; flubendazole, FLU; mebendazole, MBZ; and oxfendazole, OFZ.

**Figure 2 molecules-26-05118-f002:**
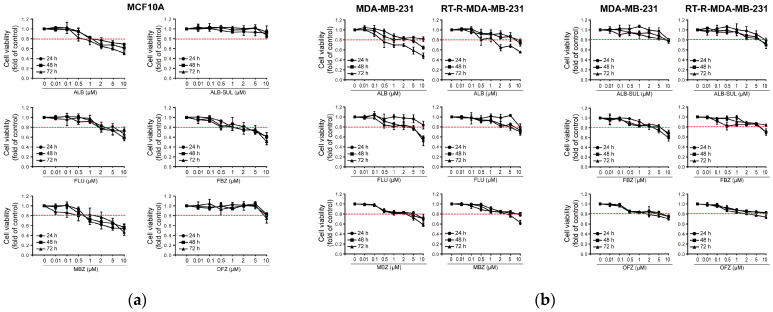
Effect of benzimidazole derivatives on the viability of normal breast epithelial cell MCF10A cells and breast cancer MDA-MB-231 and RT-R-MDA-MB-231 cells. (**a**) MCF10A, normal breast epithelial cells, or (**b**) breast cancer MDA-MB-231 and RT-R-MDA-MB-231 cells were treated with six benzimidazole derivatives (ALB, ALB-SUL, FLU, FBZ, MBZ, and OFZ) at the indicated concentrations (0.01, 0.1, 0.5, 1, 2, 5, 10 μM) for 24–72 h, and then cell viability was determined by the CCK assay. The values are expressed as the means ± SD from three independent determinations.

**Figure 3 molecules-26-05118-f003:**
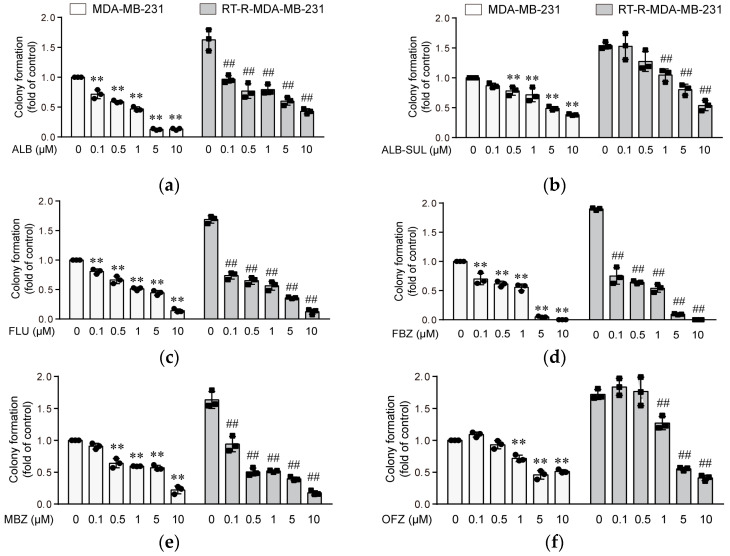
Benzimidazole derivatives ((**a**–**f**): ALB, ALB-SUL, FLU, FBZ, MBZ, and OFZ) dose-dependently reduce colony-forming ability of MDA-MB-231 and RT-R-MDA-MB-231 cells. MDA-MB-231 and RT-MDA-MB-231 cells were seeded in 6-well plates (1000 cells/well) and then treated with six benzimidazole derivatives at the indicated concentrations (0.1, 0.5, 1, 5, 10 μM). After 24 h, the culture medium was replaced with fresh medium every 2–3 days. After 10 days, the cells were fixed, stained using crystal violet, and counted. The values are expressed as the means ± SD from three independent determinations. ** *p* < 0.01 compared with the control of MDA-MB-231 cells; ^##^
*p* < 0.01 compared with control of RT-R-MDA-MB-231 cells.

**Figure 4 molecules-26-05118-f004:**
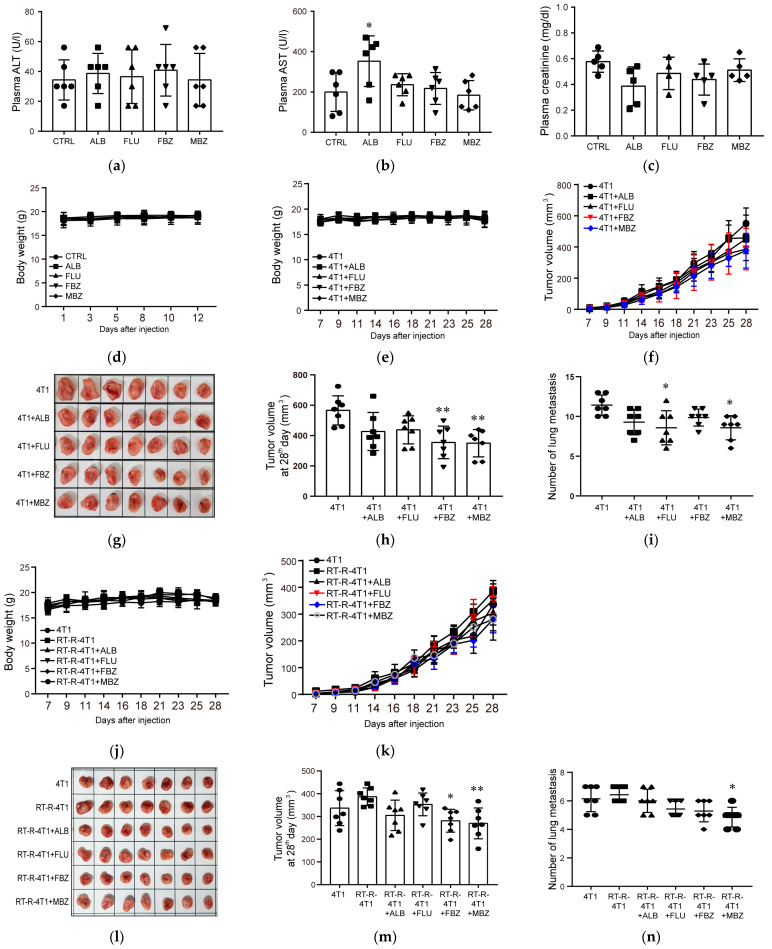
MBZ exerts the strongest anticancer effect among the benzimidazole derivatives on 4T1 and RT-R-4T1 tumor allografts in athymic nude mice. (**a**–**d**) For the safety test, the mice were divided into five groups (six mice/group) and treated with ALB, FLU, FBZ, or MBZ (10 mg/kg/mL) by oral gavage daily for 2 consecutive weeks on a schedule of 5-days-on and 2-days-off. After 2 weeks, the mice were sacrificed and plasma alanine aminotransferase (ALT) (**a**), aspartate aminotransferase (AST) (**b**), and creatinine levels (**c**) were measured. Body weights (**d**) were measured three times a week for 4 weeks. The data represent the mean ± SD (* *p* < 0.05). (**e**–**n**) 4T1 cells or RT-R-4T1 cells (5 × 10^4^ cells/100 μL) were injected subcutaneously, and when tumor volumes reached 100 mm^3^ (2 weeks after tumor injection), mice were divided into 10 groups (*n* = 7/each group): (1) 4T1 group, (2) 4T1 group + ALB, (3) 4T1 group + FLU, (4) 4T1 group + FBZ, (5) 4T1 group + MBZ, (6) RT-R-4T1 group, (7) RT-R-4T1 group + ALB, (8) RT-R-4T1 group + FLU, (9) RT-R-4T1 group + FBZ, (10) RT-R-4T1 group + MBZ for 2 consecutive weeks on a schedule of 5-days-on and 2-days-off. Body weights (**e**,**j**) and tumor volumes (**f**,**k**) were measured three times a week from 7 days after tumor cell injection. The mice were sacrificed on the 28th day after injection, and tumor volumes and (**g**,**h**,**l**,**m**) lung metastasis (**i**,**n**) were measured. The data represent the mean ± SD (* *p* < 0.05; ** *p* < 0.01).

**Figure 5 molecules-26-05118-f005:**
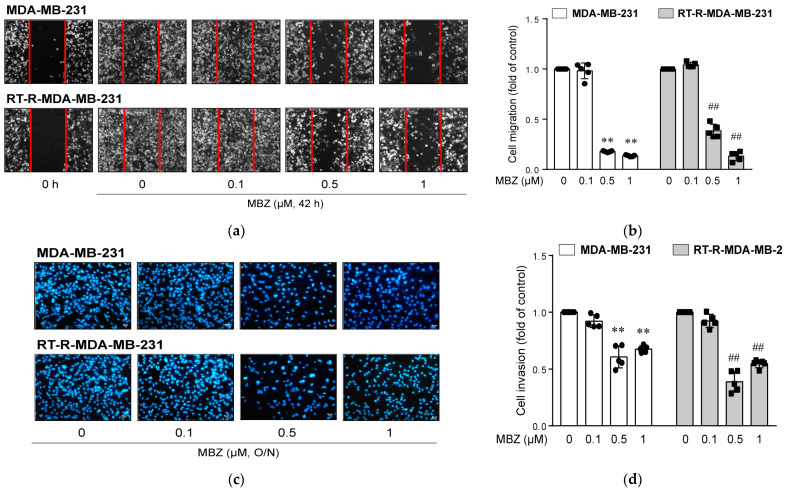
MBZ significantly reduces migration and invasion of MDA-MB-231 and RT-R- MDA-MB-231 cells at 0.5 and 1 μM. (**a**,**b**) MDA-MB-231 and RT-R- MDA-MB-231 cells were seeded in 6-well plates, and treated with MBZ (0.1, 0.5, and 1 μM), until wounds in control group without MBZ closed for 42 h. Then, the migrated cells were quantified under a microscope after wound scratching. The data represent the mean ± SD of five independent experiments. (**c**,**d**) MDA-MB-231 and RT-R- MDA-MB-231 cells were added to EC-Matrigel-coated insert wells and incubated overnight at 37 °C. The cells that had invaded across the membrane were stained with DAPI and counted in five randomly selected fields under a fluorescence microscope. The data represent the mean ± SD of five independent experiments. ** *p* < 0.01, ^##^
*p* < 0.01.

**Figure 6 molecules-26-05118-f006:**
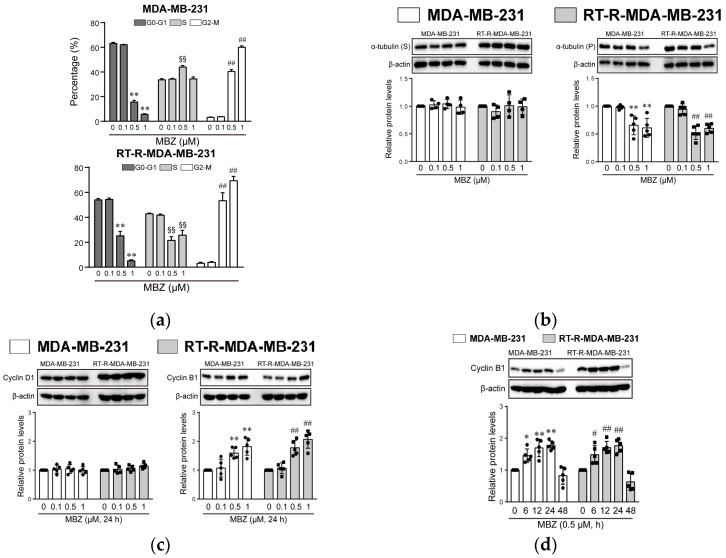
MBZ blocks cell cycle progression of both MDA-MB-231 cells and RT-R-MDA-MB-231 cells in the G2/M phase through inhibiting tubulin polymerization. (**a**–**c**) MDA-MB-231 and RT-R-MDA-MB-231 cells were treated with MBZ (0.1, 0.5, and 1 μM) for 24 h, and the (**a**) cell cycle distribution was analyzed quantitatively by flow cytometry analysis. The data represent the mean ± SD of three independent experiments. ** *p* < 0.01 compared with control in the G0-G1 phase; ^§§^
*p* < 0.01 compared with control at the S phase; ^##^
*p* < 0.01 compared with control in the G2/M phase. (**b**) α-tubulin expression and (**c**) cyclin D1 and cyclin B1 expression were analyzed by Western blot. The data represent the mean ± SD of five independent experiments. ** *p* < 0.01 compared with control of MDA-MB-231 cells; ^##^
*p* < 0.01 compared with control of RT-R-MDA-MB-231 cells. (**d**) MDA-MB-231 cells and RT-R-MDA-MB-231 cells were treated with MBZ (0.5 μM) in a time-dependent manner (6, 12, 24, and 48 h), and the cyclin D1 and cyclin B1 expression were determined by Western blotting. The data represent the mean ± SD of five independent experiments. * *p* < 0.05, ** *p* < 0.01 compared with control of MDA-MB-231 cells; ^#^
*p* < 0.05, ^##^
*p* < 0.01 (**e**,**f**) compared with control of RT-R-MDA-MB-231 cells. (**e**) Cleaved caspase-3 and pH2AX protein levels were examined in the cell lysates of cells treated with MBZ (0.1, 0.5, and 1 μM) for 24 h. The data represent the mean ± SD of five independent experiments. ** *p* < 0.01 compared with control of MDA-MB-231 cells; ^#^
*p* < 0.05, ^##^
*p* < 0.01 compared with control of RT-R-MDA-MB-231 cells.

**Figure 7 molecules-26-05118-f007:**
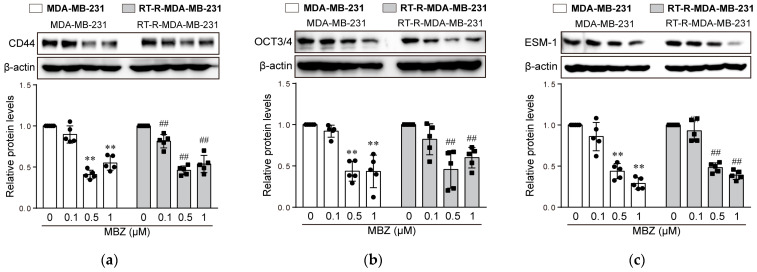
MBZ reduces the expression of cancer stem cell markers (CD44 and OCT 3/4) and ESM-1, which was enhanced in the RT-R-MDA-MB-231 cells compared to MDA-MB-231 cells. The cells were treated with the indicated concentrations of MBZ (0.1, 0.5, and 1 μM) for 24 h. The expressions of (**a**) CD44, (**b**) Oct3/4, and (**c**) ESM-1 were determined in cell lysates by Western blotting. Data are expressed as the means ± SD from five independent determinations. ** *p* < 0.01 compared with control of MDA-MB-231 cells; ^##^
*p* < 0.01 compared with control of RT-R MDA-MB-231 cells.

## Data Availability

The data presented in this study are available upon request.

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
