# Peer review of "Anticancer Effect of Benzimidazole Derivatives, Especially Mebendazole, on Triple-Negative Breast Cancer (TNBC) and Radiotherapy-Resistant TNBC In Vivo and In Vitro"

_molecules, 2021, doi:10.3390/molecules26175118_

Round 1

Reviewer 1 Report

Triple negative breast cancer (TNBC) is the most aggressive subtype of breast cancer, but an effective targeted therapy has not been well-established so far. Drug repurposing strategy will shed new light on the future directions targeting TNBC vulnerablilities with small-molecule drugs for future therapeutic purposes. In this manuscript, Jeong and the coauthors evaluated the anticancer effect on benzimidazole derivatives on TNBC and found that mebendazole (MBZ) could be potentially used as an effective anticancer agent that can overcome treatment-resistance in TNBC. However, based upon the genetic changes in the spectrum of different subsets of TNBC, individualized treatment, rather than general progress-specific treatment, has become an emerging therapeutic strategy in TNBC. A patient-centered treatment algorithm should follow the initial immunohistochemical characteristics of TNBC, so as to obtain a better objective response rate (ORR) for treatment. To achieve more understanding the molecular mechanisms of all candidates for TNBC, the classifications of TNBC from the cluster analysis of gene expression (GE) profiles need to be discussed in the background section. Despite this absence, the results reported highlight a promising scaffold against breast cancer and have scientific merit for publication in Molecules. My recommendation is to accept the manuscript after the authors have carried out the corrections as suggested below:

  1. Page 3, Line 103-104: “None of the six derivatives affected viability of TNBC MDA-MB-231 and RT-R-MDA-MB-231 cells following treatment for 24h.” The authors need to make it clear at what concentration of these six derivatives used for treatment of TNBC MDA-MB-231 and RT-R-MDA-MB-231 cells.
  2. Page 9, Line 289-291: “In this study, the doses of the drugs showing anticancer effects were 0.5 and 1 μM, which is lower than the usual dose as an anthelmintic agent.” It is not appropriate to make a comparison between the anticancer effects of benzimidazole derivatives covered in this manuscript with their anthelmintic effects. The authors need to cite the supporting data here.
  3. The authors need to recheck the formats of the cited references 1and 6.

Reviewer 2 Report

The authors in their article present research related to  the anti-cancer activity of benzimidazole derivatives toward triple negative breast cancer cells and radiotherapy resistant TNBC cells. Overall, the manuscript is clearly presented, showing interesting findings, adding to the current knowledge regarding mebendazole and its anticancer potential. I have the following questions for the authors:

-The  study focuses on the repurposing of mebendazole (MBZ) as an anti-cancer agent for the treatment of triple negative breast cancer. The anti-cancer effects of mebendazole have already been reported and some research has included triple negative breast cancer. The authors should include information in the discussion regarding previous findings.

-The authors mention in the discussion that the effective MBZ concentrations of 0.5 µM and 1 µM did not reduce the viability of breast epithelial cells nor body weight of allograft mice. However, the doses examined in vivo were 10mg/kg, please clarify this statement.

-The authors state that MBZ induced apoptosis in TNBC. However, only the levels of cleaved caspase 3 were determined. The authors could confirm apoptosis induction with Annexin V staining or an assay showing fully executed apoptosis as the authors showed that the viability of cells did not decrease significantly at the concentrations at which caspase 3 was cleaved.

-The authors mention that MBZ affect normal breast epithelial cells, however, the effects of MBZ toward these cells were higher than those toward TNBC cells. Could the authors comment on this?

Reviewer 3 Report

This is an interesting and relevant research about the effect of benzimidazole methyl carbamates on  triple-negative breast cancer as well as the potential  mechanism of cytotoxic action. In general, the relevant information is given and all the methodology and results are clearly presented. This referee is of the opinion that the manuscript is accepted in its current form.

Round 2

Reviewer 2 Report

The authors have adequately addressed my comments and questions, therefore I find the manuscript appropriate for publication.